# Anomalous X-ray diffraction studies of ion transport in K+ channels

Patricia S. Langan [1,5], Venu Gopal Vandavasi [1,6], Kevin L. Weiss [1], Pavel V. Afonine[2,3], Kamel el Omari [4], Ramona Duman[4], Armin Wagner [4] & Leighton Coates [1]

Potassium ion channels utilize a highly selective filter to rapidly transport K+ ions across cellular membranes. This selectivity filter is composed of four binding sites which display almost equal electron density in crystal structures with high potassium ion concentrations. This electron density can be interpreted to reflect a superposition of alternating potassium ion and water occupied states or as adjacent potassium ions. Here, we use single wavelength anomalous dispersion (SAD) X-ray diffraction data collected near the potassium absorption edge to show experimentally that all ion binding sites within the selectivity filter are fully occupied by K+ ions. These data support the hypothesis that potassium ion transport occurs by direct Coulomb knock-on, and provide an example of solving the phase problem by K-SAD.

[1] Neutron Scattering Division, Oak Ridge National Laboratory, 1 Bethel Valley Rd, Oak Ridge, TN 37831, USA. [2] Molecular Biophysics and Integrated Bioimaging Division, Lawrence Berkeley National Laboratory, Berkeley, CA 94720, USA. [3] Department of Physics and International Centre for Quantum and Molecular Structures, Shanghai University, Shanghai 200444, People's Republic of China. [4] Science Division, Diamond Light Source, Harwell Science and Innovation Campus, Didcot OX11 0DE, UK. [5] Present address: Lentigen Technologies, 910 Clopper Road, Gaithersburg, MD 20878, USA. [6] Present address: Department of Chemistry, Princeton University, Princeton, NJ 08540, USA. Correspondence and requests for materials should be addressed to L.C. (email: coatesl@ornl.gov)

Potassium ion (K+) channels are highly selective membrane protein pores and operate with near diffusion limited efficacy[1]. Their high selectivity and flow rates allow them to function as critical elements in electrically excitable cells, such as neurons, muscle cells, and endocrine cells[2], facilitating the use of electricity in biological organisms. The high conduction rates and ion selectivity of all K+ channels are conferred by a highly conserved selectivity filter[3] formed by a TVGYG sequence[4,5]. This selectivity filter consists of four equidistant potassium ion binding sites which are formed at the interface of four protein subunits[3,6–8] (Fig. 1). Each subunit contributes a linear backbone consisting of five or six residues with their carbonyl groups pointing inward to generate a fourfold symmetrical binding pore[3,9]. Crystal structures of the potassium channel from the soil bacteria *S. lividans* (KscA) clearly show the four discrete K+ binding sites[3,6]. However, it is not possible to determine how many K+ sites are occupied with ions at one given time, as the observed electron density at each binding site is an average value based on all possible states present within the crystal[6]. In addition, binding sites not fully occupied by K+ ions are likely to be at least partially occupied with water molecules, furthering the inability to assign electron density to a K+ alone. The correlation between the atomic displacement parameters (ADPs) and the occupancy of an atom coupled with the inability to distinguish between electron density of a K+ ion and that of a water molecule make anomalous diffraction studies necessary. To fully understand the characteristics of K+ movement in a single file through the channel, we have experimentally determined the number of K+ ions within a selectivity filter, using anomalous X-ray diffraction.

Previous anomalous scattering data has been collected from KscA in which K+ ions were replaced with thallium ions[7] (Tl+)[7], to indirectly determine the total number of K+ ions in the filter. This data indicated that the average occupancy at each site in the selectivity filter was 0.63 which inferred an occupancy of 0.53 for K+ ions, and therefore a total number of ions in the selectivity filter of two[7]. This structural observation suggested an averaging of two alternating states during which individual water molecules are translocated in between individual K+ ions across the filter[7], a commonly accepted co-translation conduction mechanism[10–12]. However, despite a multitude of studies supporting this mechanism, using a variety of in silico and experimental techniques[13–17], it has been proposed that potassium ions may translocate by direct coulomb knock-on[18–20], a scheme in which multiple adjacent binding sites in the selectivity filter are occupied by a K+ ion[20].

To experimentally determine which of these two models is correct, we conducted single wavelength anomalous diffraction (SAD) studies of K+ selective NaK2K[21,22] (NaK D66Y and N68D) at a K+ ion concentration of 100 mM. This selective NaK mutant contains the same selectivity filter as KcsA (TVGYG), and therefore four equivalent K+ binding sites[21].

## Results

**Determination of K+ occupancy in NaK2K.** The structure could be directly solved by K-SAD using the Shelx program suite[23] and Anode[24] identified strong anomalous difference peaks between 28 and 39 sigma corresponding to the K+ ions within the ion channel (Fig. 2). Using the SAD data directly, we were able to refine the occupancy of the four K+ ions within the ion channel in each of two protein molecules (subunit A and subunit B) present in the crystallographic asymmetric unit (Table 1), with the *phenix.refine* program[25]. On the top and bottom of the K+ ions in the selectivity filter are bound water molecules. The structure and experimental data have been deposited into the

protein data bank with the accession code 6DZ1. As ADPs and occupancy are closely coupled, we carefully analyzed the refined occupancies and ADPs of the K+ atoms and the oxygen atoms that are in contact with them. All of the four K+ ions in the channel refine to occupancy values close to the maximum possible value (0.25) (Table 1) with ADPs that are almost identical to the oxygen atoms located at the sides of the ion channel with which they are interacting. Finally, we used *phenix.refine* to conduct occupancy refinements on the K+ ions using 100 starting models with random occupancy and ADP values. *Phenix.refine* refines occupancies and ADP (or *B*-factors) separately at all times[25]. After refinement, the occupancy values for all the K+ ions in the channel clustered around 0.25, suggesting that all four K+ binding sites in the NaK2K selectivity filter are fully occupied with K+ ions (Fig. 3).

## Discussion

We have successfully refined potassium occupancies against anomalous diffraction data collected close to the K absorption edge. Despite the long wavelength of $\lambda = 3.35$ Å the data was of high quality allowing the crystallographic phase problem to be solved by K-SAD. The average K+ occupancy value based on the anomalous data for the A subunit and B subunit is 0.25, which is the maximum occupancy value . The high K+ ion refined occupancies at all sites in the selectivity filter indicate that all sites are fully occupied with K+ ions with a total of four K+ ions being present within the selectivity filter at a K+ ion concentration of 100 mM. These results suggest that water is not co-translocated with K+ ions, which is seemingly in disagreement with ion/water co-translocation ratios determined from earlier experiments[26,27]. However, it is in total agreement with earlier molecular dynamics simulations and crystallographic data analysis[20], suggesting that ion transport occurs via direct coulomb knock-on.

## Methods

**Protein purification**. A plasmid containing the NaK2K from *Bacillus cereus* m1550 in the pD441 vector was purchased from ATUM and transformed into *Escherichia coli* BL21 competent cells (Millipore Sigma). Briefly, NaK2K was overexpressed and purified as previously described[22]. Cultures were inoculated by scraping colonies from transformation plates into LB media, grown at 37 °C, and induced at $A_{600}$ 0.6 with 0.4 mM IPTG for 18 h at 25 °C. The cells were pelleted by centrifugation and resuspended in 5 ml lysis buffer per 1 g cells (50 mM Tris pH 7.8, 100 mM KCl), SIGMAFAST™ protease inhibitor tablets (Millipore Sigma), 1 mg ml⁻¹ of lysozyme (Calbiochem), and benzonase nuclease (Millipore Sigma). Cell resuspension was slowly stirred at room temperature for 30 min and further lysed by sonication. Cell debris was removed by centrifugation at $10,000 \times g$. NaK2K was then solubilized by incubating supernatant at room temperature for 2 h, with 40 mM Sol-grade n-Decyl-β-D-maltopyranoside n-Decyl-β-D-maltoside (DM) from Anatrace. Further debris was removed from lysate by centrifugation at $21,000 \times g$ for 30 min. Protein was purified on a Clontech labs TALON® metal affinity resin using buffers containing 4 mM DM. Protein containing fractions were pooled and the 6XHis-Tag was removed by adding 1 unit of thrombin per 1 mg of protein and incubating at room temperature for 16 h. NaK2K was then concentrated using a 30KDa Vivaspin® 20 concentrator and further purified on a Superdex 200 increase 10/300 GL column using 20 mM Tris:HCl pH 7.8, 100 mM KCl, and 4 mM Anagrade DM.

**Protein crystallization**. NaK2K protein solution was concentrated to 14 mg/ml using a 50 kDa MWCO Vivaspin® 20 concentrator (A280 of 0.55). The crystal used for long-wavelength data collection was grown in sitting drops with a final K+ concentration of 100 mM, prepared by mixing equal volumes of protein solution in buffer (20 mM Tris:HCl pH 7.8, 100 mM KCl, and 4 mM Anagrade DM) with well solution (72.5% MPD, 100 mM KCl, and 100 mM MES pH 6). The crystal was flash frozen in liquid nitrogen with no further cryoprotectant added.

**Data collection**. A 2.26 Å resolution SAD dataset was collected using X-rays with a wavelength of 3.35 Å on the long-wavelength beamline I23[28] at Diamond Light Source, UK. 360° of data were collected as an inverse-beam dataset of 20° wedges,

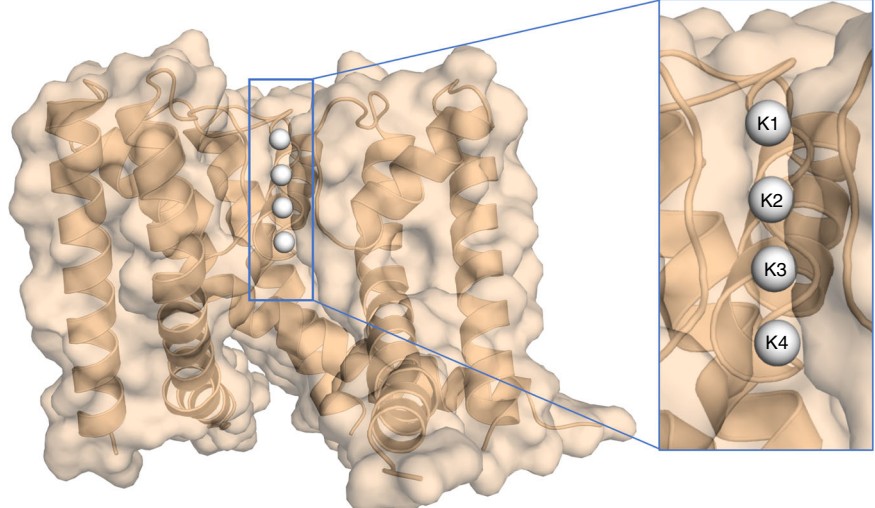

**Fig. 1** Structure of the NaK2K ion channel protein. The four integral $K^+$ binding sites within the ion channel of a $K^+$ selective NaK2K (NaK D66Y and N68D) at a $K^+$ ion concentration of 100 mM are shown as white spheres. For clarity one of the four subunits that make up the ion channel has been removed

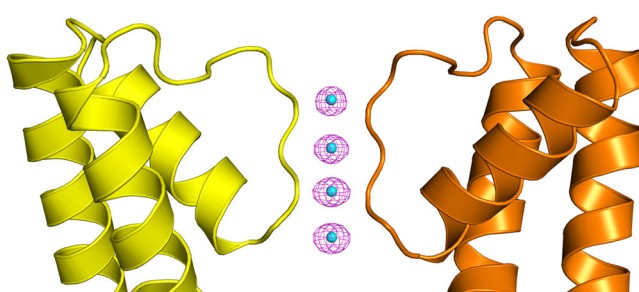

**Fig. 2** Anomalous difference map. The anomalous difference Fourier map contoured at $8\sigma$ is shown as a magenta mesh for subunit A. Strong anomalous difference peaks corresponding to the $K^+$ ions (cyan spheres) are present within the selectivity filter of the ion channel. For clarity only two of the four subunits that make up the ion channel are shown

**Table 1 Anomalous occupancy refinement values of $K^+$ ions within the ion channel**

| Binding site | Residue ID | Anomalous refined occupancy [100 mM $K^+$] | Anomalous peak height ($\sigma$) |
|---|---|---|---|
| $A_1$ | $K_1$ | 0.23 | 28.50 |
| $A_2$ | $K_2$ | 0.25 | 30.03 |
| $A_3$ | $K_3$ | 0.28 | 35.67 |
| $A_4$ | $K_4$ | 0.27 | 36.74 |
| $B_1$ | $K_1$ | 0.22 | 36.66 |
| $B_2$ | $K_2$ | 0.24 | 36.74 |
| $B_3$ | $K_3$ | 0.26 | 39.37 |
| $B_4$ | $K_4$ | 0.30 | 38.63 |

Sites $A_1$ to $A_4$ occur within the A chain while sites $B_1$ to $B_4$ occur within the B chain. Each $K^+$ ion within the selectivity filter is shared between four unit cells and thus has a maximum occupancy value of 0.25. Occupancy values >0.25 arise due to the correlation between atomic displacement parameter (B-factor) and occupancy

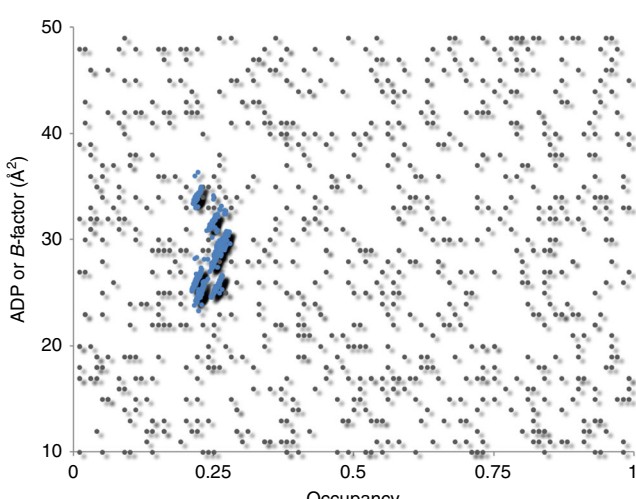

**Fig. 3** Results of occupancy refinement with differing starting points. One hundred starting models with initial occupancies and ADP values for $K^+$ ions randomly drawn from [0–1] to [10–50] ranges for each of eight $K^+$ ions per structure (gray dots). The refined $K^+$ occupancy values for each $K^+$ ion (blue dots) cluster around 0.25

with an exposure of 0.1 s per 0.1° rotation, and beam attenuated to 20%. A second 360° dataset was collected from the same crystal, using a different crystal orientation, by opening the goniometer kappa axis by 10° and the phi axis by 70°.

At the time of the experiment, the fluorescence detector at the beamline had not yet been fully commissioned. Therefore, rather than optimizing the anomalous contribution based on an energy scan across the absorption edge and subsequent quantitative analysis of the scan to determine f″, we tuned the X-ray energy to 3700 eV, 91.6 eV above the tabulated potassium K absorption edge (3608.4 eV). This is far enough in energy from the near edge region (XANES) characterized by large fluctuations of f″ due to resonance effects within the specific coordination sphere of the potassium atoms. While the absolute value of f″ is slightly reduced further away from the absorption energy, this approach allows using the theoretical approximation of 3.9 electrons[29] for sufficiently accurate anomalous occupancy refinements. In summary, we collected a complete SAD dataset at a wavelength of 3.35 Å, which is close to the K absorption edge (3.44 Å), resulting in a very strong anomalous signal (Table 2) from a theoretical anomalous contribution f″ of 3.9 electrons from K[29].

As water molecules do not generate a significant anomalous signal at the X-ray wavelength used for data collection, this method is able to experimentally discriminate between a superposition of $K^+$ ions and water molecules and a fully occupied $K^+$ ion in each binding site. This data was reduced with XDS[30] and scaled with XSCALE[31]. Analysis of the data showed strong anomalous scattering of $CC_{1/2}$ anomalous = 59% overall and $CC_{1/2}$ anomalous = 33% at $D_{min}$ = 2.26 Å. The source of this signal could only be the $K^+$ ions, since the protein sequence lacks any sulfur-containing amino acids. The known structure of the selective NaK2K mutant[21] was used as a starting model and refinement was conducted using the Phenix[32] suite of programs, while the coot[33] molecular graphics program was used to model the structure in between rounds of refinement. The Ramachandran plot

**Table 2 Data collection and refinement statistics**

|  | NaK2K |
|---|---|
| *Data collection* |  |
| Space group | I4 |
| Cell dimensions |  |
| *a, b, c* (Å) | 68.07, 68.07, 89.38 |
| *α, β, γ* (°) | 90, 90, 90 |
| Resolution (Å) | 54.15–2.26 (2.33–2.26) |
| $R_{merge}$ | 3.70 (12.90) |
| $I/\sigma I$ | 33.40 (10.70) |
| Completeness (%) | 97.80 (93.40) |
| Redundancy | 8.20 (6.30) |
|  |  |
| *Refinement* |  |
| Resolution (Å) | 54.15–2.26 |
| No. reflections | 18417 |
| $R_{work}/R_{free}$ | 0.1782/0.2112 |
| No. atoms |  |
| Protein | 1482 |
| Ligand/ion | 10 |
| Water | 28 |
| *B*-factors |  |
| Protein | 39.18 |
| Ligand/ion | 38.35 |
| Water | 43.41 |
|  |  |
| R.m.s. deviations |  |
| Bond lengths (Å) | 0.014 |
| Bond angles (°) | 1.09 |

of the final model contained 98.91% of residues in favored conformations with no outliers. The data collection and refinement statistics are shown in Table 2. The figures within this manuscript were made using the pymol program[34]. The program Xtriage from the Phenix suite[32] was used to check for data for signs of crystal twinning, with no twin fraction being detected.

## Data availability

The refined protein structure and associated structure factors have been deposited in the protein data bank with the accession code 6DZ1. All other data supporting the findings of this study are available from the corresponding author on reasonable request.

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

## Acknowledgements

Research at the Spallation Neutron Source (SNS) at ORNL was sponsored by the Scientific User Facilities Division, Office of Basic Energy Sciences, U.S. Department of Energy. The Office of Biological and Environmental Research supported research at the Center for Structural Molecular Biology (CSMB) at ORNL using facilities supported by the Scientific User Facilities Division, Office of Basic Energy Sciences, U.S. Department of Energy. L.C. thanks Youxing Jiang and Nam Nguyen from UT Southwestern for kind help and advice on the expression and crystallization of the protein. P.V.A. acknowledges support by the NIH (grant GM063210), the Phenix Industrial Consortium and the US Department of Energy under Contract No. DE-AC02-05CH11231.

## Author contributions

L.C. conceived and supervised research. P.S.L., K.L.W., and V.G.V. expressed, purified, and crystallized the protein. A.W., R.D., and K.E.O. collected and reduced the data. L.C., R.D., and P.V.A. refined the data. L.C. wrote the paper with help from all co-authors.

## Additional information

**Competing interests:** The authors declare no competing interests.

