## [Peer Review File · Nature Communications]

Referee's report on Nature Comms submission

My report is in three parts; on the article narrative, on the PDB Validation report and on the underpinning data (the diffraction data set and protein model).

A. The article narrative

This is an important study. The article is well written. It looks to me like a scientifically exciting article worthy of inclusion in Nature Comms. An article which reinvestigated this structure from the prior data in the PDB was:-

Ion permeation in K^+ channels occurs by direct Coulomb knock-on David A. Köpfer et al

Science 17 Oct 2014: Vol. 346, Issue 6207, pp. 352-355 DOI: 10.1126/science.1254840

This article should be cited in the introduction to the new study.

It is important to stress that this submission to NatureComms is also a methodology tour de force exemplifying the importance of the Diamond Light Source diffractometer in vacuo for accessing successfully the potassium K edge. This is then a second major reason for my recommending acceptance for Nature Comms.

B. The PDB Validation report

I see no particular problems highlighted in the PDB's report.

C. The underpinning data

The Fo-Fc difference Fourier map, that I calculated in Refmac from the mtz file provided by the authors, provides a commendably very short list of peaks that might lead though to improved model details or need commenting on in the Supplementary details. The top 3 peaks are given in this screenshot:-

Peak 1 suggests that the B factor for this water molecule (number 82) is overestimated at 62 \AA^2 . Likewise peak 2 for that water (number 83) at a B factor of 43 \AA^2 .

Peak 3 looks too close to place a water, I agree.

The anomalous difference Fourier map is outstandingly excellent. One has to simply marvel at it!:-

Coordinate Series A:-

Coordinate Series B:-

I commend that the above two figures are added to the article ideally in the main section or if needs be for article length reasons in the supplementary.

The anomalous difference Fourier map above was calculated in Phenix.

The Fo-Fc peaks list calculated now by Phenix is:-

What is the peak 3 in the above screen shot? A bound water? [Ie since there is no anom peak on it.] It looks as if it may be functionally interesting. A comment is needed in the article.

Water 83 identified in the Refmac Fo-Fc as having a too high a B factor actually has an anomalous difference peak on it:-

So does water 12:-

Water 34 also, although this anom peak is quite small:-

These three bound waters need reassignment to ions, their identity as a cation or anion according to their charged neighbour.

So my scrutiny of the underpinning data suggests some improvements to the protein model are possible and maybe also appropriate comments made in the article or supplementary details, as indicated above.

Reviewer #2 (Remarks to the Author):

This paper presents an x-ray structure of a K selective mutant of NaK. The main focus of the paper is K occupancy in the selectivity filter. A K SAD experiment is used to determine absolute occupancy. The data are processed with the phenix.refine program to give an occupancy of ~ 0.25 ($\times 4$) = 1.

Some technical questions:

How was f'' of 'around 4 electrons' determined? Was this measured directly? If so, please provide data on it.

While the randomization of ADP and occupancy values shows convergence to occupancy ~ 0.25 , and the authors state the importance of decoupling ADP and occupancy, what was really done to do so? (i.e. what's inside phenix.refine?)

Reference 22 used TI data from reference 7 to show occupancy = 1 by similarly running a program (Shelx), but without a deliberate effort to decouple occupancy and ADP. But in ref 7 the authors carried out alternate cycles of ADP (what was called B factor) and occupancy refinement (always fixing one while refining the other) and they found TI occupancy = 0.63. Did you consider such an approach.

As I see it, there are 2 key technical complications in occupancy refinement. The first is getting the number of f'' electrons right (ideally by measuring it) and the second is decoupling occupancy and ADP to the best of one's ability (unfortunately they are inconveniently correlated). This paper does not describe how either of these things were done.

I raise a final question for the authors to think about, and perhaps comment on.

How does a conduction mechanism with 4 fully occupied K sites account for the well-documented thermodynamic coupling of water and K movement through a K channel (i.e. streaming potential)?

We thank the reviewers for taking time to help us improve our manuscript. We have thought carefully about all the points raised, below is a point by point explanation of how we responded to each reviewer comment. Author comments are in black text with our response following in blue text.

Reviewer #1 (Remarks to the Author):

Ion permeation in K⁺ channels occurs by direct Coulomb knock-on David A. Köpfer et al Science 17 Oct 2014: Vol. 346, Issue 6207, pp. 352-355 DOI: 10.1126/science.1254840
This article should be cited in the introduction to the new study.

We have added this reference into the introduction of the revised article.

The anomalous difference Fourier map is outstandingly excellent. One has to simply marvel at it!:-I commend that the above two figures are added to the article ideally in the main section or if needs be for article length reasons in the supplementary.

We have added a figure into the manuscript (Figure 2) which shows the anomalous difference Fourier map for ion channel.

" **Figure 2:** The anomalous difference Fourier map contoured at 8σ is shown as a magenta mesh for subunit A. Strong anomalous difference peaks corresponding to the K⁺ ions (cyan spheres) are present within the selectivity filter of the ion channel. For clarity only two of the four subunits that make up the ion channel are shown."

What is the peak 3 in the above screen shot? A bound water? [Ie since there is no anom peak on it.] It looks as if it may be functionally interesting. A comment is needed in the article.
Water.

We have added a water molecule at this position into the structure and we added this sentence into the manuscript.

"On the top and bottom of the K⁺ ions in the selectivity filter are bound water molecules, the structure and experimental data has deposited into the protein data bank with the accession code 6DZ1."

These three bound waters need reassignment to ions, their identity as a cation or anion according to their charged neighbour.

The three bound water molecules have been reassigned as K⁺ ions in the structure.

Reviewer #2 (Remarks to the Author):

This paper presents an x-ray structure of a K selective mutant of NaK. The main focus of the paper is K occupancy in the selectivity filter. A K SAD experiment is used to determine absolute occupancy. The data are processed with the phenix.refine program to give an occupancy of $\sim 0.25 \times 4 = 1$.

Some technical questions:

How was f'' of 'around 4 electrons' determined? Was this measured directly? If so, please provide data on it.

At the time of the experiment the beamline fluorescence detector had not yet been fully commissioned, so a quantitative measurement based on a XANES scan was not possible. The X-ray energy was chosen 90 eV above the theoretical absorption edge to avoid any potential white line effects in the XANES region. At 3.7 keV (3.35 Å wavelength) the theoretical value for f'' is 3.9 e-, in the text we state "around 4 e-" indicating that there is some uncertainty about the absolute value. Hence, we are confident that avoiding the XANES region gives us only a small uncertainty / discrepancy from between theory and experiment, not affecting the conclusions made in the manuscript. Source: <http://skuld.bmsc.washington.edu/scatter/data/K.dat>

Based on:

DT Cromer and D Liberman (1970), J. Chem. Phys. 53, 1891.

We have added the following sentences into the manuscript to address this point.

"We collected a complete dataset on the long wavelength beamline I23(24) at the Diamond Light Source synchrotron at 3.35 Å. This wavelength is close to the K absorption edge (3.44 Å), resulting in a very strong anomalous signal (Table 1) from a theoretical anomalous contribution f'' of 3.9 electrons from K(25)."

And we added these sentences into the supporting information

" At the time of the experiment the fluorescence detector at the beamline had not yet been fully commissioned. Therefore, rather than optimizing the anomalous contribution based on an energy scan across the absorption edge and subsequent quantitative analysis of the scan to determine f'' , we tuned the X-ray energy to 3700 eV, 91.6 eV above the tabulated potassium K absorption edge (3608.4 eV). This is far enough in energy from the near edge region (XANES) characterized by large fluctuations of f'' due to resonance effects within the specific coordination sphere of the potassium atoms. While the absolute value of f'' is slightly reduced further away from the absorption energy, this approach allows using the theoretical approximation of 3.9 electrons(25) for sufficiently accurate anomalous occupancy refinements."

While the randomization of ADP and occupancy values shows convergence to occupancy ~ 0.25 , and the authors state the importance of decoupling ADP and occupancy, what was really done to do so? (i.e. what's inside phenix.refine?)

phenix.refine refines occupancies and ADP (or B-factors) separately at all times, as well as coordinates. This is documented in phenix.refine paper (<http://journals.iucr.org/d/issues/2012/04/00/ba5180/ba5180.pdf>) a refinement run consists of macro-cycles, with each macro-cycles consisting of independent steps of refining coordinates, occupancies, B-factors and other atomic and non-atomic model parameters.

Reference 22 used TI data from reference 7 to show occupancy = 1 by similarly running a program (Shelx), but without a deliberate effort to decouple occupancy and ADP. But in ref 7 the authors carried out alternate cycles of ADP (what was called B factor) and occupancy refinement (always fixing one while refining the other) and they found TI occupancy = 0.63. Did you consider such an approach.

phenix.refine refines occupancies and ADP (or B-factors) separately at all times, as well as coordinates. As documented in phenix.refine paper (<http://journals.iucr.org/d/issues/2012/04/00/ba5180/ba5180.pdf>) a refinement run consists of macro-cycles, with each macro-cycles consisting of independent steps of refining coordinates, occupancies, B-factors and other atomic and non-atomic model parameters. So the approach outlined above seems to be an inherent feature of *phenix.refine*. We have added the following sentence into the manuscript to clarify this point.

" Phenix.refine refines occupancies and ADP (or B-factors) separately at all times(30)."

As I see it, there are 2 key technical complications in occupancy refinement. The first is getting the number of f'' electrons right (ideally by measuring it) and the second is decoupling occupancy and ADP to the best of one's ability (unfortunately they are inconveniently correlated). This paper does not describe how either of these things were done.

We have added extra sentences into the manuscript detailing how the number of f'' electrons were determined in response to an earlier comment.

"We collected a complete dataset on the long wavelength beamline I23(24) at the Diamond Light Source synchrotron at 3.35 Å. This wavelength is close to the K absorption edge (3.44 Å), resulting in a very strong anomalous signal (Table 1) from a theoretical anomalous contribution f'' of 3.9 electrons from K(25)."

The decoupling of occupancy and ADP refinement is an inherent feature of *phenix.refine* which refines occupancies and ADP (or B-factors) separately at all times, as well as coordinates. We have added the following sentence into the manuscript to clarify this point.

" Phenix.refine refines occupancies and ADP (or B-factors) separately at all times(30)."

I raise a final question for the authors to think about, and perhaps comment on. How does a conduction mechanism with 4 fully occupied K sites account for the well-documented thermodynamic coupling of water and K movement through a K channel (i.e. streaming potential)?

This is an interesting point and not an easy one to resolve using the data within our manuscript alone. We have added the following sentences into our manuscript that acknowledge this fact.

"Our results suggest that water is not co-translocated with K ions which is seemingly in disagreement with ion/water co-translocation ratios determined from earlier experiments(31, 32) . However, it is total agreement with earlier molecular dynamics simulations and crystallographic data analysis(22)."